# The Association between Hip Joint Morphology and Posterior Wall Fracture: Analysis of Radiologic Parameters in Computed Tomography

**DOI:** 10.3390/jpm13091406

**Published:** 2023-09-20

**Authors:** Han Soul Kim, Ki Uk Mun, Chul-Ho Kim

**Affiliations:** 1Department of Orthopedic Surgery, Gachon University Gil Medical Center, Incheon 21556, Republic of Korea; 2Department of Orthopaedic Surgery, Asan Medical Centre, University of Ulsan College of Medicine, Seoul 05505, Republic of Korea

**Keywords:** fracture, acetabulum, hip joint, diagnostic imaging, computed tomography

## Abstract

Although numerous radiologic parameters of abnormal hip joint morphology are utilized in practice, studies on the relation of these parameters to acetabular fractures are limited. This study hypothesized that certain morphological features of hip joints are associated with acetabular posterior wall (PW) fracture patterns and aimed to identify morphological characteristics predictive of acetabular PW fracture. The records of 107 consecutive patients, who were diagnosed with acetabular fractures in a level I trauma center from August 2017 to April 2021, were initially reviewed. After excluding patients who lacked proper radiographic evaluation and had previous surgery or concomitant injury on the ipsilateral lower limb, a total of 99 patients were analyzed to investigate the morphological characteristics of the hip joint, measured in computed tomography, associated with acetabular posterior wall fracture. We included patient demographics, acetabular index (AI), sharp angle, acetabular depth-to-width ratio (AD/WR), center-edge angle (CEA), head–neck offset ratio (HNOR), acetabular head index (AHI), anterior acetabular sector angle (AASA), posterior acetabular sector angle (PASA), and acetabular version angle (AVA) in the univariate and multivariate analyses. The injury mechanism (*p* = 0.001) and AD/WR (*p* = 0.021) were predictors of PW fracture in the univariate analysis. In the multivariable analysis, injury mechanism (*p* = 0.011), AI (coefficient B = 0.320; Exp (B) = 1.377; *p* = 0.017), and AD/WR (coefficient B = 33.047; Exp (B) = 2.250 × 10^14^; *p* = 0.028) were significant predictors of PW fracture. This study highlights the importance of morphological factors, such as a larger AI and AD/WR, that may influence joint stress distribution, resulting in acetabular PW fracture. Understanding these pathomechanisms may protect the hip joint and prevent future injuries through the early identification and treatment of pathological conditions.

## 1. Introduction

Acetabular posterior wall (PW) fractures account for approximately one third of all acetabular fractures [1,2,3]. Acetabular PW fractures usually result from high-energy trauma in young patients and cause poor outcomes, thereby leaving them with functional and financial burdens many years after the index trauma [4,5,6,7]. Of patients with acetabular fractures, 85% are accompanied by dislocation, which may lead to catastrophic sequelae, such as trauma-related femoral head osteonecrosis or recurrent dislocation [5,8]. Moreover, a recent increase in elderly patients with acetabular fractures and an aging population makes the treatment of acetabular PW fracture even more challenging [9,10,11].

Identifying the fracture pattern using Letournel’s classification system is critical to achieving a successful treatment outcome of acetabular fractures [12,13,14,15]. The specific fracture patterns are correlated with the injury mechanism, the vector of injury force application, innominate bone anatomy, and its mechanical properties [12,13]. Acetabular PW fractures mostly result from a “dashboard” injury, during which a force is transmitted to the PW of the flexed hip joint through the femoral axis [16]. Understanding the anatomy and pathomechanism of a certain disease leads to its prevention, treatment, and improvements in treatment outcome. The dysplastic hip has been related to femoroacetabular impingement (FAI) and hip joint osteoarthritis [17,18,19]. However, the relation between abnormal hip joint morphology and acetabular fractures has not been identified despite the high incidence and clinical significance of acetabular PW fractures.

Although numerous radiologic parameters of abnormal hip joint morphology are utilized in practice, studies on the relation of these parameters to acetabular fractures are limited. This study hypothesized that certain morphological features of the hip joint are associated with acetabular PW fracture patterns and aimed to identify morphological characteristics predictive of acetabular PW fracture.

## 2. Materials and Methods

### 2.1. Patient Selection

The records of 107 consecutive patients, who were diagnosed with acetabular fracture in a level I trauma center from August 2017 to April 2021, were reviewed. The study was approved by our Institutional Review Board with a waiver for the need to provide written informed consent. Data collection was performed following the relevant guidelines and regulations of the committee.

All acetabular fractures, diagnosed using three-dimensional computed tomography (3D-CT) images, were included in the study to avoid misinterpretation of fracture classification. Patients who (i) lacked 3D-CT evaluation, (ii) had a combined fracture in the ipsilateral lower limb, and (iii) underwent ipsilateral hip joint or pelvic bone surgery prior to injury were excluded from the study. A total of 99 patients were included in the study (Figure 1).

### 2.2. Fracture Classification and Measurement of Morphological Features

The pelvic anterior–posterior, inlet, outlet, and lateral views were gathered and examined to classify acetabular fractures according to Letournel’s method [15]. Matched 3D-CT images were double-checked for precise acetabular fracture pattern identification. Patients were divided into two groups for analyses: the case group with PW fractures and the control group with other acetabular fractures. 

The following nine radiologic parameters were collected to evaluate the morphological features of the acetabulum: (1) acetabular index (AI), (2) sharp angle, (3) acetabular depth-to-width ratio (AD/WR), (4) center-edge angle (CEA), (5) head–neck offset ratio (HNOR), (6) acetabular head index (AHI), (7) anterior acetabular sector angle (AASA), (8) posterior acetabular sector angle (PASA), and (9) acetabular version angle (AVA). The AI, sharp angle, AD/WR, CEA, and AHI were measured in the coronal plane of the hip center in 3D-CT [20]. The AASA, PASA, HNOR, and AVA were measured on the axial plane of the hip center [20,21,22]. All radiologic parameters were first measured on the injured hip, when possible. Measurements were performed on the contralateral hip if fracture displacement was >3 mm. All measurements were performed by two orthopedic surgeons who specialized in hip surgeries and were then averaged for final analysis. The measurement details of radiologic parameters are described in Figure 2.

### 2.3. Statistical Analysis

Patient sex, age, injury mechanism, and radiologic parameters were compared between the control and case groups. The independent *t*-test or Mann–Whitney U test was used for continuous variables, whereas the chi-square test or Fisher’s exact test was used to evaluate categorical variables, after verifying the assumption of normal data distribution.

For the primary outcome of this study, univariate and multivariate binary logistic regression analyses were conducted to find predictors of isolated acetabular PW fracture. Univariate and multivariable binary logistic regression analyses were performed on acetabular fractures with and without PW involvement. The acetabular fractures with PW involvement included isolated PW fracture, posterior wall and column (PW + PC) fracture, and transverse and PW (Transverse + PW) fracture. All parameters which the authors presumed to be clinically associated with the incidence of PW fracture or its involvement were included in the final analyses despite the statistical insignificance in the univariate analysis. 

All statistical analyses were performed using PASW Statistics version 18.0 (IBM Corp., Armonk, NY, USA). A *p* value of <0.05 was considered significant.

## 3. Results

### 3.1. Patient Demographics

Of the 99 patients with acetabular fracture, 27 showed isolated acetabular PW fracture. The most common type of acetabular fracture was PW (27 of 99, 27.3%) and isolated anterior wall fracture (22 of 99, 22.2%), followed by anterior column fracture (21 of 99, 21.2%). Two cases of PW + PC fracture and one case of Transverse + PW fracture were identified. The majority of the included patients were male (81 of 99, 81.8%), and the mean age was 50.1 ± 17.0 (16–71) years. The most common cause of acetabular fracture was falling from a height in the overall patient group (33 of 99, 33.3%) as well as in the control group (32 of 72, 44.4%). However, the most common cause of acetabular PW fracture was a motor vehicle accident (16 of 27, 59.3%). The radiologic parameters that characterize hip joint morphology and other details are shown in Table 1 and Table 2.

### 3.2. The Radiologic Risk Factor Analysis for Isolated PW Fracture

The univariate and multivariable logistic regression analyses results for potential risk factors associated with isolated acetabular PW fracture are presented in Table 3. The univariate analysis revealed a significant association of injury mechanism (*p* = 0.001) and AD/WR (*p* = 0.021) with isolated PW fracture. With multivariable analysis, injury mechanism (*p* = 0.011), acetabular index (coefficient B = 0.320; Exp (B) = 1.377; *p* = 0.017), and acetabular depth-to-width ratio (coefficient B = 33.047; Exp (B) = 2.250 × 10^14^; *p* = 0.028) were significant predictors of isolated PW fracture (Table 3).

### 3.3. The Radiologic Risk Factor Analysis for Acetabular Fractures with PW Involvement

The univariate and multivariable logistic regression analyses results for potential risk factors associated with acetabular fractures with PW involvement are presented in Table 4. The univariate analysis revealed a significant association of injury mechanism (*p* < 0.001) and AD/WR (*p* = 0.036) with acetabular fractures involving the PW component. With multivariable analysis, injury mechanism (*p* = 0.006) and acetabular index (coefficient B = 0.255; Exp (B) = 1.291; *p* = 0.035) were significantly associated with acetabular fractures involving the PW component (Table 4).

## 4. Discussion

This study primarily revealed that a larger AI and AD/WR are radiologic risk factors for acetabular PW fracture and a larger AI is the sole radiologic risk factor for any acetabular fractures with PW involvement.

Acetabular PW fractures are the most common type of acetabular fractures. An epidemiologic study in 2014 revealed that acetabular PW fractures accounted for 32% in the United States and 30% in China [3]. A review article by Kelly et al. revealed that the most common fracture types were associated with both column fractures (22.3%) and PW fractures (20.9%), followed by transverse + PW fractures (16.3%) [2]. In this study, PW fractures accounted for 27.3% of all acetabular fractures, which was comparable to the aforementioned studies. We observed a noticeable change in the mean patient age from previously published reports. The updated systematic review on acetabular fractures noted that the mean patient age has risen from 38.6 to 45.2 years compared to the previous systematic review published in 2004 [2,23]. Our results suggest that the mean patient age may have risen even higher, to 50.1, which could be attributed to the high percentage of patients aged over 60 years (30/99, 30.3%). Several authors have called for more attention to the increasing incidence of acetabular fractures among geriatric patients because these patients with increased comorbidities and osteoporosis pose a greater challenge to surgeons [9,10,11,24].

Multiple studies have linked hip joint morphology to FAI and osteoarthritis [17,18,19,25]. Regarding traumatic hip joint injury, few studies have identified proximal femoral geometry, such as neck-shaft angle and femoral anteversion, as a predisposing factor for proximal femoral fractures [26,27,28]. Two groups of scholars have indicated that hip dysplasia and FAI play a role in posterior hip dislocation [29,30]. Traditionally, FAI occurs in combination with morphological factors and activity level; that is, pathological repetitive contact accumulates stress concentration in certain hip joint areas until the symptom presents itself [31]. This study aimed to identify the morphological factors that may predispose the hip joint to posterior wall fractures when combined with a traumatic event. To the best of our knowledge, only one group has attempted to correlate acetabular morphology with acetabular fractures, although no relevant acetabular geometry was identified [24]. Our study identified larger AI and AD/WR as risk factors for acetabular PW fractures, and a larger AI was the sole risk factor for acetabular fractures containing PW fragments. These results suggest that less anterosuperior femoral head coverage and a deeper hip socket are predisposing factors for acetabular PW fractures.

The AI is one of the most widely used radiologic parameters to evaluate acetabular coverage [32,33]. The larger AI in our study indicates less anterior wall and more dysplastic hip coverage. A reduced load transfer area causes increased contact pressure, leading to accumulated labrum microdamage and eventual tearing, in a biomechanically unstable joint of a dysplastic hip [31]. Nearly 90% of dysplastic hips accompany labral tear [34]. An intact labrum provides equal stress distribution across the hip joint; however, a torn labrum results in increased stress concentration to the articular cartilage, which may dramatically increase under higher stress conditions, such as high-energy falls or motor vehicle accidents, compared to normal weight-bearing. Moreover, dysplastic hips are known to have cam deformities in 42% of cases, which cause impingements during hip flexion in the anterosuperior quadrant, femoral head posterior subluxation, and subsequently increased contact pressure at the acetabular PW [28]. We propose that less anterosuperior acetabular coverage in a dysplastic hip leads to increased stress concentration on the posterior compart of the hip joint, predisposing the acetabulum to fracture under trauma.

The AD/WR is also one of the representative radiologic parameters of acetabular depth [35,36]. The larger the AD/WR, the deeper the hip socket. We found that a larger AD/WR is a significant predictor of acetabular PW fracture. In deep socket or pincer deformities, femoral head overcoverage limits the range of hip motion, and greater force is transmitted to the acetabular rim as the femoral neck impinges [25]. We believe that high-energy force applied to a deeper hip joint exposes the acetabular rim to a greater risk of fragmentation as the femoral neck abuts the rim. The injury mechanism is one of the well-known predictors of the fracture pattern of the acetabulum. Our results showed that injury from motor vehicle accidents as a driver accounted for nearly 60% of all injury mechanisms in acetabular PW fractures. When we performed logistic regression analyses to eliminate the compounding effect of injury mechanism as a risk factor, the AI and AD/WR were still significantly associated with acetabular PW fracture. Therefore, we believe this study highlights the importance of morphological factors, mainly a larger AI and AD/WR, that may influence joint stress distribution, resulting in acetabular posterior wall fracture. 

The current study has several limitations. First, it is retrospective in nature and the number of cases documented is relatively small. However, considering that acetabular fractures are rare injuries in heterogenous patient groups compared to other orthopedic trauma, and considering that this is the first study to successfully identify the relationship between acetabular geometry and acetabular fracture patterns, our study has shown great value in understanding the morphological risk factors of acetabular PW fractures. Second, although we attempted to measure radiologic parameters on the injured side, 69 cases were measured on the contralateral side, and gathering side-specific information was hindered by large fracture gaps and deformities. However, as Gebre et al. suggested, both sides showed a high correlation, indicating that the contralateral side is valid for radiologic parameter measurements [24]. In addition, the indicators of acetabular overcoverage or pincer deformities cannot solely be represented by the AI; that is, a CEA of >40°, an alpha angle of >55°, and an AVA of <15° may all indicate overcoverage or pincer deformities [37]. Similarly, parameters of acetabular dysplasia include not only the AD/WR but also a CEA of <20°, a sharp angle of >45°, an AI of >14°, an AHI of <75%, an AASA of <50°, and a PASA of <90° [20]. We analyzed a wide variety of radiologic parameters for hip joint morphology; however, we were able to identify only the AI and AD/WR as predictors of acetabular PW fracture. High-energy trauma and the relative position of the femoral head against the acetabulum are the two well-known predictors of acetabular fracture patterns. This study suggests that underlying pathologic conditions, such as impingement and instability, can predispose an innominate bone to acetabular PW fracture, which brings debilitating consequences to patient function and quality of life. In the era of ageing populations and a growing incidence of fragility acetabular fractures, treating such pathologic conditions before repetitive microdamage accumulates in the hip joint of geriatric patients may prevent future fractures.

## 5. Conclusions

This study highlights the importance of morphological factors, such as a larger AI and AD/WR, that may influence joint stress distribution, resulting in acetabular posterior wall fracture. Understanding these pathomechanisms may protect the hip joint and prevent future injuries through the early identification and treatment of pathological conditions. 

## Figures and Tables

**Figure 1 jpm-13-01406-f001:**
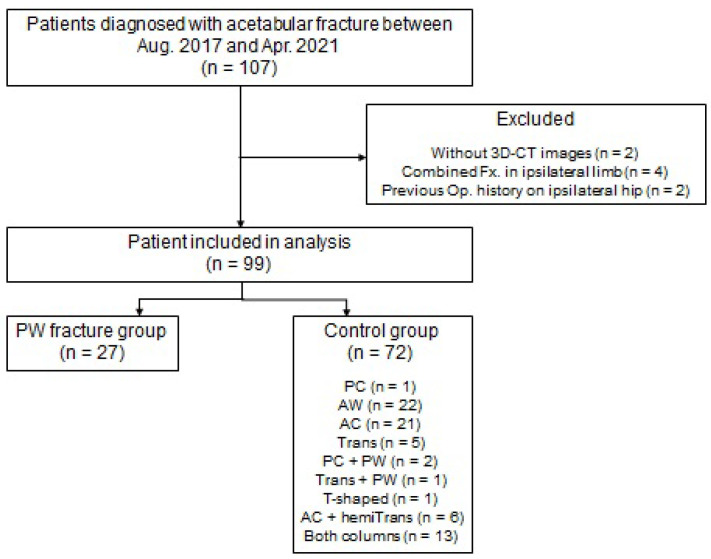
The flowchart of the patient selection process—3D-CT: three-dimensional computed tomography; Fx: fracture; Op: operation; PW: posterial wall; PC: posterior column; AW: anterior wall; AC: anterior column; Trans: transverse; hemiTrans: hemitransverse.

**Figure 2 jpm-13-01406-f002:**
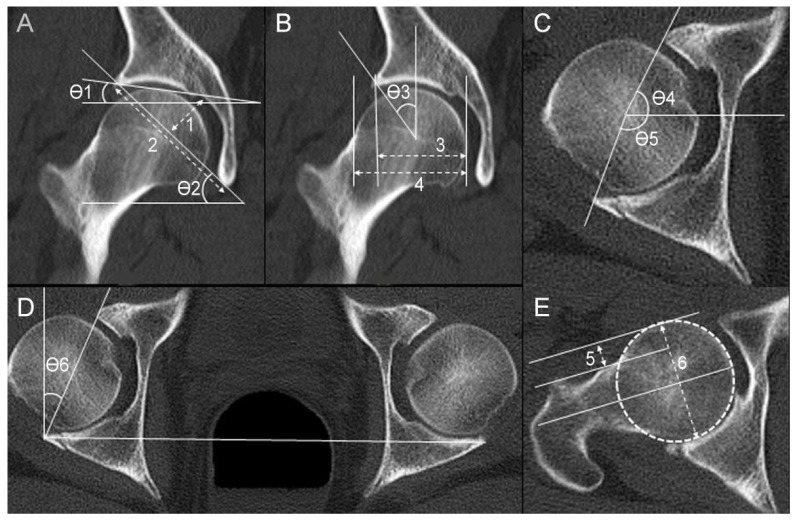
The measurements of radiologic hip joint parameters are shown in a 55-year-old male patient with a right acetabular posterior wall fracture. (**A**) ϴ1: acetabular index; ϴ2: sharp angle; (1/2): acetabular depth-to-width ratio. (**B**) ϴ3: center-edge angle; (3/4): acetabular head index. (**C**) ϴ4: anterior acetabular sector angle; ϴ5: posterior acetabular sector angle. (**D**) ϴ6: acetabular version angle. (**E**) (5/6): head–neck offset ratio.

**Table 1 jpm-13-01406-t001:** Patient demographics and injury mechanism.

	Total (n = 99)	PW (n = 27)	Control (n = 72)	*p*-Value
Male sex	81 (81.8%)	24 (88.9%)	57 (79.2%)	0.383
Age (year)	50.1 ± 17.0 (16–71)	45.3 ± 17.0 (16–71)	51.9 ± 17.6 (13–90)	0.095
Injury mechanism				<0.001
Slip down	6 (6.1%)	0 (0%)	6 (8.3%)	
Fall from height	33 (33.3%)	1 (3.7%)	32 (44.4%)	
MVA: driver	21 (21.2%)	16 (59.3%)	5 (6.9%)	
MVA: passenger	7 (7.1%)	2 (7.4%)	5 (6.9%)	
Pedestrian	11 (11.1%)	1 (3.7%)	10 (13.9%)	
Bicycle	5 (5.1%)	1 (3.7%)	4 (5.6%)	
Motorcycle	11 (11.1%)	5 (18.5%)	6 (8.3%)	
Crushing injury	5 (5.1%)	1 (3.7%)	4 (5.6%)	

PW: posterior wall fracture group; MVA: motor vehicle accident.

**Table 2 jpm-13-01406-t002:** Radiologic parameters of hip joint morphology.

Radiologic Parameters	Total (n = 99)	PW (n = 27)	Control (n = 72)	*p*-Value
AI (°)	8.67 ± 4.33 (0.73–20.51)	8.88 ± 4.71 (0.73–20.51)	8.59 ± 4.21 (0.82–20.32)	0.770
Sharp angle (°)	39.57 ± 4.19 (25.25–49.08)	39.76 ± 4.07 (30.71–49.08)	39.50 ± 4.25 (25.25–46.92)	0.788
AD/WR	0.27 ± 0.03 (0.21–0.41)	0.29 ± 0.04 (0.24–0.41)	0.27 ± 0.03 (0.21–0.34)	0.047
CEA (°)	32.02 ± 6.71 (14.47–48.23)	32.29 ± 6.50 (18.92–43.54)	31.91 ± 6.83 (14.47–48.23)	0.805
AHI	0.81 ± 0.07 (0.65–1.17)	0.82 ± 0.09 (0.69–1.17)	0.81 ± 0.06 (0.65–0.95)	0.797
AASA (°)	61.46 ± 7.93 (28.40–77.78)	61.63 ± 8.48 (47.73–77.24)	61.40 ± 7.77 (28.40–77.78)	0.900
PASA (°)	91.86 ± 8.04 (74.74–116.74)	92.60 ± 6.95 (79.21–108.76)	91.59 ± 8.44 (74.74–116.74)	0.580
HNOR	0.20 ± 0.06 (0.09–0.34)	0.20 ± 0.06 (0.11–0.30)	0.19 ± 0.06 (0.09–0.34)	0.847
AVA (°)	15.79 ± 5.30 (3.79–29.07)	16.28 ± 5.18 (5.75–25.53)	15.61 ± 5.37 (3.79–29.07)	0.575

PW: posterior wall fracture group; AI: acetabular index; AD/WR: acetabular depth-to-width ratio; CEA: center-edge angle; AHI: acetabular head index; AASA: anterior acetabular sector angle; PASA: posterior acetabular sector angle; HNOR: head–neck offset ratio; AVA: acetabular version angle.

**Table 3 jpm-13-01406-t003:** Results of univariate and multivariable binary logistic regression analyses for predictors of isolated posterior wall fracture.

Characteristic	Univariate Analyses			Multivariable Analyses		
	B	Exp (B)	*p*-Value	B	Exp (B)	*p*-Value
Sex (Male)	−0.744	0.475	0.272	−0.917	0.400	0.444
Age (Year)	−0.022	0.978	0.097	−0.046	0.103	0.955
Injury mechanism	N/A	N/A	0.001	N/A	N/A	0.011
AI (°)	0.015	1.016	0.767	0.320	1.377	0.017
Sharp angle (°)	0.015	1.015	0.785	−0.216	0.806	0.073
AD/WR	15.878	7.868 × 10^6^	0.021	33.047	2.25 × 10^14^	0.028
CEA (°)	0.008	1.008	0.803	0.050	1.051	0.661
AHI	0.800	2.224	0.795	−10.080	<0.001	0.405
AASA (°)	0.004	1.004	0.899	0.141	1.151	0.168
PASA (°)	0.016	1.016	0.576	0.051	1.053	0.527
HNOR	0.756	2.130	0.846	−1.032	0.356	0.876
AVA (°)	0.024	1.025	0.571	0.326	0.057	0.057

AI: acetabular index; AD/WR: acetabular depth-to-width ratio; CEA: center-edge angle; AHI: acetabular head index; AASA: anterior acetabular sector angle; PASA: posterior acetabular sector angle; HNOR: head–neck offset ratio; AVA: acetabular version angle.

**Table 4 jpm-13-01406-t004:** Results of univariate and multivariable logistic regression analyses for predictors of posterior wall fracture involvement.

Characteristic	Univariate Analyses			Multivariable Analyses		
	B	Exp(B)	*p*-Value	B	Exp(B)	*p*-Value
Sex (Male)	−0.916	0.400	0.175	−1.286	0.276	0.261
Age (Year)	−0.022	0.978	0.092	−0.046	0.955	0.104
Injury mechanism	N/A	N/A	<0.001	N/A	N/A	0.006
AI (°)	0.013	1.014	0.790	0.255	1.291	0.035
Sharp angle (°)	0.042	1.043	0.434	−0.108	0.897	0.356
AD/WR	13.815	9.99 × 10^5^	0.036	22.645	6.836 × 10^9^	0.112
CEA (°)	0.012	1.012	0.726	0.066	1.069	0.519
AHI	0.872	2.392	0.770	−6.012	0.002	0.581
AASA (°)	0.007	1.007	0.794	0.041	1.041	0.675
PASA (°)	0.019	1.019	0.491	0.094	1.099	0.287
HNOR	0.316	1.372	0.933	−2.079	0.125	0.742
AVA (°)	0.011	1.011	0.789	0.154	1.167	0.339

AI: acetabular index; AD/WR: acetabular depth-to-width ratio; CEA: center-edge angle; AHI: acetabular head index; AASA: anterior acetabular sector angle; PASA: posterior acetabular sector angle; HNOR: head–neck offset ratio; AVA: acetabular version angle.

## Data Availability

The datasets generated and analyzed during the current study are not publicly available since they contain potentially identificatory information for each patient, but they are available from the corresponding author on reasonable request.

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
