# Peer review of "The Association between Hip Joint Morphology and Posterior Wall Fracture: Analysis of Radiologic Parameters in Computed Tomography"

_jpm, 2023, doi:10.3390/jpm13091406_

Round 1
Reviewer 1 Report
The authors provide a well-presented paper investigating morphological patterns of the hip joint and their association with acetabular posterior wall fractures. Overall, the paper is well presented, but some minor changes would improve readability. Of note, some clarifying details are missing from the methods which are noted below.
Title:
1. The phrase "prediction" implies causality. Consider changing to "associations"
Abstract:
2. PW is undefined before the first use of acronym. Use in full at the first instance.
3. It would be useful to add to the abstract some details about exclusion criteria to maintain consistency with the methods. On initial read, the n's did not appear to match.
4. The statistical analysis method is underdescribed in the abstract, in particular what constituted the groups for analysis. More detail would be useful.
Methods
5. A number of undefined acronyms are used in Figure 1. I suggest writing out in full or providing the full in caption.
Results
6. For readability, I would recommend separating out the two regression analyses (isolated PW vs PW involvement endpoints). In the current format, it's a bit hard to immediately catch the differences between the two analyses. Clearly separating into subsections and flagging the differences in definition between the two groups would be useful.
7. How were the regression models fitted? Were all variables maintained in the final model, or was some kind of selection procedure used?
Author Response
We appreciate your taking the time to review our paper and suggest constructive comments. Responses to the comments are attached as word file.

Reviewer 2 Report
In the manuscript, the authors report their experience of hip joint morphological features associated with acetabular posterior wall (PW) fracture patterns. They aimed to identify morphological characteristics predictive of acetabular PW fracture. There were included and reviewed records of 99 consecutive patients who were diagnosed with acetabular fractures in a level I trauma center from August 2017 to April 2021. The authors used univariate and multivariable binary logistic regression analyses to detect morphological characteristics of the hip joint, measured in computed tomography, associated with acetabular PW fracture. Results of this study suggest that the mean patient age may have risen even higher to 50.1, which could be attributed to the high percentage of patients over 60 years. More prominent AI and AD/WR were identified as risk factors for acetabular PW fractures, and more extensive AI was the sole risk factor for acetabular fractures containing PW fragments. The results showed that less anterosuperior femoral head coverage and deeper hip socket are predisposing factors for acetabular PW fractures. The more extensive AI indicated less anterior wall and more dysplastic hip coverage. The morphological factors, such as more extensive AI and AD/WR, may influence joint stress distribution, resulting in acetabular posterior wall fracture. The authors described details and underlined the role of the pathomechanisms that may protect the hip joint and prevent future injuries by early identification and treatment of pathological conditions. It highlights the role of this study and can offer new directions for studies in the future.
Some details for improvement of this article I have underlined in the comments. Please find the attached document describing a few minor issues noted in the manuscript.

Author Response

(The authors gave the same response as above.)
